# Cross-Cultural Adaptation and Validation of the Arabic Version of Musculoskeletal Health Questionnaire (MSK-HQ-Ar)

**DOI:** 10.3390/ijerph17145168

**Published:** 2020-07-17

**Authors:** Fahad Saad Algarni, Abdulmajeed Nasser Alotaibi, Abdulrahman Mohammed Altowaijri, Hana Al-Sobayel

**Affiliations:** 1Department of Rehabilitation Sciences, College of Applied Medical Sciences, King Saud University, Riyadh 11433, Saudi Arabia; hsobayel@ksu.edu.sa; 2Ministry of Health, Children Hospital, Taif 26514, Saudi Arabia; 437105492@student.ksu.edu.sa; 3Department of Rehabilitation Medicine, King Saud University Medical City, King Saud University, Riyadh 11461, Saudi Arabia; abaltowaijri@ksu.edu.sa

**Keywords:** Arabic translation, cross-cultural adaptation, musculoskeletal disorders, musculoskeletal health questionnaire, psychometric properties, validity, reliability

## Abstract

Background: Musculoskeletal disorders (MSD) affect millions of people worldwide. Musculoskeletal Health Questionnaire (MSK-HQ) is a valid and reliable tool to assess the health of patients with MSD. However, this scale is not available in the Arabic language. The purpose of this study was to translate and cross-culturally adapt the Musculoskeletal Health Questionnaire (MSK-HQ) into Arabic (MSK-HQ-Ar) and evaluate its validity and reliability among participants with MSD. Methods: This cross-sectional study used guidelines from the International Society for Pharmacoeconomics and Outcomes Research (ISPOR) to translate as well as validate the psychometric properties of MSK-HQ-Ar. Patients with MSD (*n* = 149) living in Taif participated in the study. Cronbach’s alpha and the intraclass correlation coefficient (ICC) were used to assess internal consistency and test-retest reliability of MSK-HQ-Ar respectively. Spearman’s correlation was used to assess the correlation between MSK-HQ-Ar and the European quality of life five-dimension, five-level scale (EQ-5D-5L). Results: Out of 149 participants, 119 completed the MSK-HQ-Ar twice. The scale showed good internal consistency, Cronbach’s alpha (0.88), and reliability (ICC 0.92–0.95). A strong association was found with the EQ-5D-5L scores. Conclusion: The adapted MSK-HQ-Arabic version revealed acceptable psychometric properties and is a valid and reliable outcome measure to assess MSK health among Arabic speaking patients with MSD.

## 1. Introduction

Musculoskeletal disorders (MSDs) constitute relatively common conditions across all sociodemographic populations and can be defined as ‘any injury to the human support system of bones, joints, and soft tissue that can occur from a single event or a cumulative trauma’ [1]. These disorders have serious physical and psychological implications on the health of millions of people worldwide, and affect their families and work as well [1,2,3,4]. Approximately 30% of American adults complain of joint pain, swelling, or limitations at any given time [5]. In fact, MSDs represent one of the most common causes of both long-term pain and physical disability, leading to work disability, absenteeism as well as diminished productivity. Moreover, these conditions have an enormous negative impact on the economic burden of health care and welfare systems [1,6,7]. Generally, patients with MSDs are assessed based on the years lived with disability (YLD) measure, a concept that the World Health Organization defines as the “sum of years of potential life lost due to premature mortality and the years of productive life lost due to disability”. Based on this measure, MSDs contribute significantly towards YLDs by causing 40% of all chronic conditions and 54% of all long-term disabilities [8]. Previous studies found high prevalence of MSDs in Saudi Arabia [9,10,11] and demonstrated several challenges to the management of different MSDs [12]. Saudi Arabia’s health care system includes both primary and secondary care settings, including public as well as private health care centres. These centres include a wide array of medical professionals involved to treat patients with MSDs and may have an increased risk of care fragmentation across different health care settings and a lack of consistency of care along the healthcare pathway.

Among various interventions designed to treat MSDs, physical therapy embodies one of the more effective treatments for easing the symptoms of MSDs [13]. Traditionally, the physical therapy assessment focuses on intensity and duration of MSK pain, range of motion, and muscle strength. Consequently, the use of standardised measures to assess patient’s daily functional activities in clinical practice is being neglected [14,15], thus indicating the need for valid and reliable outcome measures. In recent years, a wide range of Patient-Reported Outcome Measures (PROMs) have been developed and tested, with the aim of increasing patients’ involvement in the provision of care [16]. A few validated measures were used to assess MSK health and the MSK-HQ is one of the valid and reliable scale. It comprises a patient-reported generic PROM that captures a wide range of MSK conditions involving different pathways of care. This outcome measure was developed by the Arthritis Research UK Primary Care Sciences Research Centre in collaboration with Keele University and the University of Oxford [17]. The final version of the MSK-HQ includes 14 items covering 12 items. A comparison of the MSK-HQ with other outcome measures (OMs) reveals that this measure is relatively simple, acceptable, easy to understand, score, and interpret, and also empowers patients, who can monitor themselves at home. It is a valid and reliable measure and also strongly correlates with both the European quality of life five-dimension, five-level scale (EQ-5D-5L) and the six-item Keele MSK-PROM. However, this scale has not been translated into Arabic. Culturally adapting an existing health questionnaire constitutes a faster and more economical approach than developing a new tool. In addition, the availability of such outcome measures in different languages may facilitate future comparisons among cross-cultural populations [18]. Hence, this study aimed to translate and culturally adapt the Musculoskeletal Health Questionnaire (MSK-HQ) into Arabic language (MSK-HQ-Ar) and analyse its psychometric properties among patients with MSD in Saudi Arabia.

## 2. Methods

### 2.1. Study Design

This study was conducted in two phases: The first phase involved the translation and cultural adaptation of the MSK-HQ from English to Arabic. Secondly, the psychometric properties of Arabic version of the MSK-HQ including internal consistency and reliability was assessed. The adaptation process was performed according to the International Society for Pharmacoeconomics and Outcomes Research (ISPOR) guidelines [19], which comprised widely accepted standards for the translation and adaptation of the outcome measures. The Ethical Review Board of the College of Applied Medical Science, King Saud University (CAMS139a-3839) approved the study.

#### 2.1.1. Phase 1: The Translation and Cross-Cultural Adaptation of the MSK-HQ-Ar

The MSK-HQ was translated into Arabic based on the ISPOR guidelines. These guidelines have received approval from regulatory agencies such as the Food and Drug Administration (FDA) and the European Medicines Agency (EMEA) as well as the original author [20]. The translation process included several steps (Figure 1).

In the first step, the forward translation step, two experienced translators independently adapted and translated the original English version of the MSK-HQ into Arabic. The two translators were native Arabic-speaking physical therapy and rehabilitation health professionals; thus, they were able to successfully produce colloquial translations of the original English version. In the second step, the forward translation reconciliation step, the two forward translated versions (F1 and F2) were compared with each other to resolve any differences between translations. After resolving any issues, the translations were merged into a single forward translation by taking the common terms in the individual speech habits and preferences. Any differences between translations were resolved by a key in-country investigator and an independent reviewer. They were then further modified by the project manager to finally produce a reconciled version of the translation (MSK-HQ-Ar) that accurately represented the concepts in the original English version as well as the cultural nuances that are meaningful in Saudi Arabia.

In the next stage, to maintain the quality of the translated version, the MSK-HQ-Ar version was back-translated into the English language via the backward translation step. Next, two backward translations of the reconciled version of the MSK-HQ-Ar into English were developed (B1 and B2). Then in the fourth step, the back-translation review, the B1 and B2 versions were compared with the original English version and the discrepancies were highlighted by key in-country investigators. Most of the discrepancies were encountered in Items 12 and 13, which were corrected appropriately in the pre-final version. For instance, under “introduction”, the item “describe you” was replaced by “describe your status”. Additionally, for Items 1–11, “not at all” was changed to “never happened”, and in Item 9, “trouble” was replaced by “difficulty”. These discrepancies were resolved, and finally, a pre-final Arabic version was developed and evaluated via pilot testing.

#### 2.1.2. Pilot Testing

The pre-final version of the MSK-HQ-Ar was formatted to match the layout of the original MSK-HQ. Pilot testing was conducted to evaluate the MSK-HQ-Ar to ensure proper quality control. A total of fourteen (*n* = 14) Saudi patients with MSDs in the upper extremity, lower extremity, and spinal regions were recruited for the study. All the participants were provided with the pre-final Arabic version of the questionnaire. The mean time to complete the questionnaire was 4.5 min (SD ± 1.4). Out of the fourteen participants, the majority were male *n* = 9 (64%) and *n* = 5 (36%) were females. The mean age of the participants was 36.7 years old (SD ± 12.4 years), mean BMI was 24.9 (SD ± 4). Three out of the fourteen patients showed a little difficulty with some words like stiffness during the night, daily routine, and fatigue. The comments and suggested alternatives made by the participants were reviewed by the in-country investigator to highlight any discrepancies in the meaning or terminology used. The queries and suggestions were forwarded to the project manager for discussion and finally an Arabic version of the MSK-HQ with the best possible translation was developed.

#### 2.1.3. Phase 2: Testing the Psychometric Properties of the MSK-HQ-Ar

A cross-sectional study was conducted to evaluate the psychometric properties including internal consistency, test-retest reliability, and construct validity of the final version of the MSK-HQ-Ar among patients diagnosed with MSD, living in the Taif region of Saudi Arabia. The convenient sample (*n* = 153) was selected from physical therapy clinics at the King Faisal Medical City (KFMC) and King Abdulaziz Specialist Hospital (KAASH) from their general practitioner (GP). The study excluded participants that lacked the ability to read or speak Arabic and suffered from comorbidities such as neurological, craniological, pulmonary, or cognitive conditions. Finally, four out of the 153 participants that failed to meet the eligibility criteria were excluded from the study, thus resulting in a final total of 149 participants. Initial sample size calculation revealed an estimated sample size of *n* = 118, with a minimum acceptable intraclass correlation coefficient (ICC) of ρ = 0.70, and an expected ICC of ρ1 = 0.80 [21]. However, a total of 153 participants were included to deal with dropouts, if any. The participants were 84 males and 65 females between the ages of 18 and 65 years. All participants were primarily diagnosed as having MSDs and were referred to physical therapy clinics.

#### 2.1.4. Data Collection

Each participant was given a clear and detailed explanation about the purpose of the study and participation was voluntary and confidential. A signed written informed consent was obtained from each participant before administrating the questionnaires. At baseline, the set of all questionnaires were completed by each participant including the demographic data sheet, the MSK-HQ-Ar, and the EQ-5D-5L. The average time to complete all the questionnaires was 12–18 min. For test-retest reliability, a sample (*n* = 119) completed the MSK-HQ-Ar twice. At the second visit, each participant of the subgroup completed the Global Rating of Change Scale (GRC) before the treatment session to identify participants with stable symptoms to be involved in retest reliability assessment. Then, a copy of the MSK-HQ-Ar was administered to them. The second assessment was taken after two weeks to avoid memorisation and any recall bias.

## 3. Instruments

The MSK-HQ-Ar constitutes a generic OM that measures the holistic impact of an MSK condition on a person’s health regardless of the pain location or care position on the clinical pathway. This questionnaire contains 14 items that cover 12 domains, including pain severity, physical function, work interference, social interference, sleep, fatigue, emotional health, physical activity, independence, understanding, confidence to self-manage, and overall impact. Each of the items provide response options based on a five-point Likert scale [17].

The EQ-5D-5L constitutes a generic OM that evaluates patients’ health status. This tool assesses five domains of health: mobility, self-care, usual activities, pain/discomfort, and anxiety/depression [22,23]. In addition, the EQ-5D-5L includes five levels of intensity for each assessed factor: difficulties, minor problems, fair problems, acute problems, and extreme problems. An Arabic version of the EQ-5D-5L tool was used in this study [24]. The GRC consists of a self-rated OM that assesses patients’ perception of changes in their postintervention health status, was also used [25]. Hence, this measure was used in our study to quantify a patient’s improvement over time and assess the patient’s current health status.

## 4. Statistical Analysis

The Statistical Package for the Social Sciences (SPSS) software (IBM SPSS Statistics for Windows, Version 23.0; IBM Corp., Armonk, NY, USA) conducted the data analysis. The basic features of the participants’ demographic data were analysed using a number of descriptive statistics, including frequencies, percentages, means and/or standard deviations. In this study, the floor and ceiling effects of the translated scale were assessed by computing the number of participants, as percentages, who had the highest score or the lowest score of the scale. The internal consistency of the OM was examined using Cronbach’s alpha, with the acceptable alpha level being between 0.70 and 0.95. The test-retest reliability of the MSK-HQ-Ar was assessed using the intraclass correlation coefficient (ICC) with a 95% confidence interval (95% CI). An acceptable level of ICC is 0.70 or higher. Construct validity was evaluated using Spearman’s rank correlation analysis to determine the association between the MSK-HQ-Ar and the EQ-5D-3L Arabic version. The level of significance was set at *p* ≤ 0.05.

## 5. Results

### 5.1. Demographic Characteristics

Among the 149 participants, 84 (56.4%) were male and 65 (43.6%) were female. All study participants were between 18 and 65 years of age, with a mean age of 43.6 years old (SD ± 13.6 years). Most of the participants fell into the 40- to 49-year-old age group (28%), followed by those in the 50–59 age bracket (20%). Participants had a mean body mass index (BMI) of 28.5 kg/m^2^ (SD ± 5.6 kg/m^2^). Around (39%) had a BMI of 30 or greater, 36.3% had BMI ranged from 25 to 29.9, while 19.4% were in the 18.6–24.9 range. Only 5% of the participants had a BMI of 18.5 or lower. The education of all participants ranged from primary to postgraduate level. Fewer than half of the participants (45%) had a university degree, while 24% of the participants had received a secondary school education, and only 7% of the participants completed intermediate school. The employment status of the participants varied, with 40% of the participants being employed and 36% unemployed, and 24% of the participants were either students or retired. Lastly, most of the study participants were married (77%) (Table 1).

Participants reported MSDs in different body regions. In general, most of the respondents complained of lower extremity MSDs (46%), followed by spinal MSDs (36%) and upper extremity MSDs (18%). Participants also reported musculoskeletal pain in the knee joint (41%), lumbar region (30%), shoulder (16%), cervical region (5.4%), and other areas (6%), including the elbow, wrist, hip, and ankle (Table 2).

### 5.2. MSK-HQ-Ar Scores

#### 5.2.1. Floor and Ceiling Effect

An analysis of the test instruments revealed the completeness of items in all measures. The Shapiro–Wilk test determined the normal distribution of MSK-HQ-Ar total scores, as shown in Table 3, revealed a distribution with a significance level of 0.226 (*p* > 0.05), skewness value of −0.122 (0.199), and Kurtosis value of −0.819 (0.395). Specifically, the MSK-HQ-Ar total score demonstrated a normal distribution among the participants scoring across the entire range from 5 to 53. This finding implies the lack of floor or ceiling effects, with the highest score of 53 representing only 2% of the total scores and the lowest score, 5, representing 0.7% of the total scores. In addition, none of the participants reported the highest or lowest MSK-HQ-Ar scores. The same pattern of results was observed for the EQ-5D-5L.

#### 5.2.2. Internal Consistency and Test-Retest Reliability

Cronbach’s alpha (α) was used to assess the internal consistency of the MSK-HQ-Ar and was 0.88, which is acceptable and good. The corrected item-to-total correlations ranged from 0.41 to 0.67, with the highest correlation for Item 6, “Work/daily routine,” and the lowest correlation for Item 12, “Understanding of your condition and any current treatment.” The deletion of an item from the MSK-HQ-Ar failed to significantly change the alpha level, as values ranged from 0.86 to 0.88 when deleting an item at baseline (Table 4).

A test-retest reliability analysis was performed on 119 stable participants. This analysis underwent examination by using the ICC two-way random-effect model for participants who completed the MSK-HQ-Ar questionnaire two times within a two-week recall period. The ICC level for the total number of items ranged between 0.92 and 0.95. Thus, the total ICC of 0.94 indicated an excellent agreement of test-retest reliability, with a standard error of measurement (SEM) of 2.46 and a minimal detectable change at 95% CI (MDC_95_) of 6.83 (Table 5). 

### 5.3. Construct Validity

The Spearman’s correlation coefficient was used to assess the correlation between the total scores of MSK-HQ-Ar and the EQ-5D-5L. The analysis showed a significantly strong correlation with the EQ-5D-5L (Rho = 0.711, *p* < 0.001). 

The Arabic version of the MSK-HQ may be used in a wide range of MSDs along different pathways of healthcare. Statistical analyses showed that the Arabic version of MSK-HQ demonstrated strong reliability and validity for use among the Arabic population. The process of adaptation and evaluation revealed a lack of major problems and missing data, which indicated a strong acceptance of MSK-HQ-Ar.

## 6. Discussion

This study aimed to translate the MSK-HQ into the Arabic language and test its psychometric properties, internal consistency, test–retest reliability, and construct validity. The current study findings suggest that he MSK-HQ-Ar is a reliable and valid tool for assessing the health outcomes in patients with MSD in Saudi Arabia. In this study, the investigators thoroughly read, discussed, cross-checked, and proofread all the translation versions and managed to reach consensus decisions on all discrepancies, thus ensuring equivalence between the translated Arabic MSK-HQ version and the original MSK-HQ. In this study, the majority of the recruited patients rated the adapted version of the MSK-HQ as simple, clear, and understandable with very few issues. In comparison to the EQ-5D-5L and the Short Form Health survey (SF-36), the MSK-HQ-Ar requires approximately two additional minutes to complete. The original MSK-HQ exhibited a Flesch reading ease test score of 65.9, which indicates its ease of understanding for 13- to 15-year-old students. In comparison to the EQ-5D-5L, which received a Flesch score of 61.3, the MSK-HQ has greater readability [17].

The MSK-HQ-Ar exhibited strong internal consistency and excellent test-retest reliability in this study. The Cronbach’s alpha of the Arabic version, was good at α = 0.88, and is similar score to the internal consistency of the original English version of the scale [17]. The Italian version (MSK-HQ-I) had a slightly lower Cronbach’s alpha (0.871) [26]. Secondly, the correlation of MSK-HQ-I with the EQ-5D-5L was 0.674, while the correlation of MSK-HQ-Ar was 0.711 [26].

The current study also showed that the calculated value reflects the homogeneity of the MSK-HQ-Ar, as the items within the OM correlate with each other when measuring the same trait [27]. In particular, Item 6 correlated most strongly to the total score. This item, entitled “interference with work/daily routine,” had an alpha of 0.67, followed by the Items 14, 7, and 9, which possessed alphas ranging from 0.675 to 0.627. This pattern resembled the results of the original study, where Item 6 exhibited the strongest correlation, with a 0.76 alpha. Furthermore, the weakest correlated item, Item 12, “understanding of the condition,” had an alpha of 0.41. This score also demonstrated a consistent pattern with the original study, where the weakest item, Item 12, correlated with alpha −0.04 and 0.32 (17). In 2018, Norton et al. reported the highest alpha level, α = 0.93, for the MSK-HQ. The high score of this value resulted from the internal consistency being tested on a homogenous sample of patients with inflammatory arthritis [28].

This study assessed the stability of the MSK-HQ-Ar by taking measurements of patients who showed stable symptoms at two different time periods. The recall period of up to two weeks represented a sufficient time period to circumvent the memorization effect while avoiding biological changes that occur over longer time periods [27]. In addition, the study examined the MSK-HQ-Ar stability in terms of test-retest reliability. This study demonstrated an excellent value of ICC = 0.94, almost similar to the values reported for Italian version (ICC = 0.963) [26], but a higher value than the original English version [17,28]. Furthermore, the study assessed the absolute reliability or measurement error resulting from repeated measures in terms of SEM and MDC, both of which support the clinical importance in evaluating the real effects of interventions and change over time. In this study, the SEM was calculated at 2.46 and the MDC_95_ was 6.8, which indicates the minimal value constituting a real change in a single subject. As no earlier studies reported the calculated SEM and MDC, these values provided an added advantage to our study. 

The results of the current study showed a significantly strong correlation with EQ-5D-5L, having a Rho of 0.72 and *p <* 0.001. These findings demonstrated consistency with the original study, which also reported a strong correlation with EQ-5D-5L, having a Rho of 0.82 in the same cohort of physical therapy participants. Correlation with another similar outcome measure with a correlation coefficient of 0.4 or greater has been suggested to show good construct validity of the scale [29]. Norton et al.’s (2019) validation study results showed a strong correlation with EQ-5D (r = 0.80), which demonstrated consistency with the study results. The overall mean of MSK-HQ-Ar score in this study was 32.29 (10.42), while it was 28.62 (9.61) for MSK-HQ [17] and 37.39 (9.36) for MSK-HQ-I [26].

### Study Limitations

The major limitation of the present study involved its lack of an EQ-5D-5L value for the Saudi population or the population of the Middle East. Furthermore, the current study used a small convenient sample. Hence, future studies should recruit and employ a large and random sample from different Arabic regions. This measure could further validate and enhance the findings of the present study. 

## 7. Conclusions

In conclusion, the MSK-HQ-Ar is a reliable and valid tool for assessing the impact of MSDs on patients’ health along the healthcare pathway. This study also found a significant correlation between the MSK-HQ-Ar and EQ-5D-5L. More extensive research using this tool is necessary to study the health of MSD patients residing in Saudi Arabia and other Arabic speaking countries. These psychometric properties of the MSK-HQ-Ar suggest the usefulness of this tool in the clinical setting.

## Figures and Tables

**Figure 1 ijerph-17-05168-f001:**
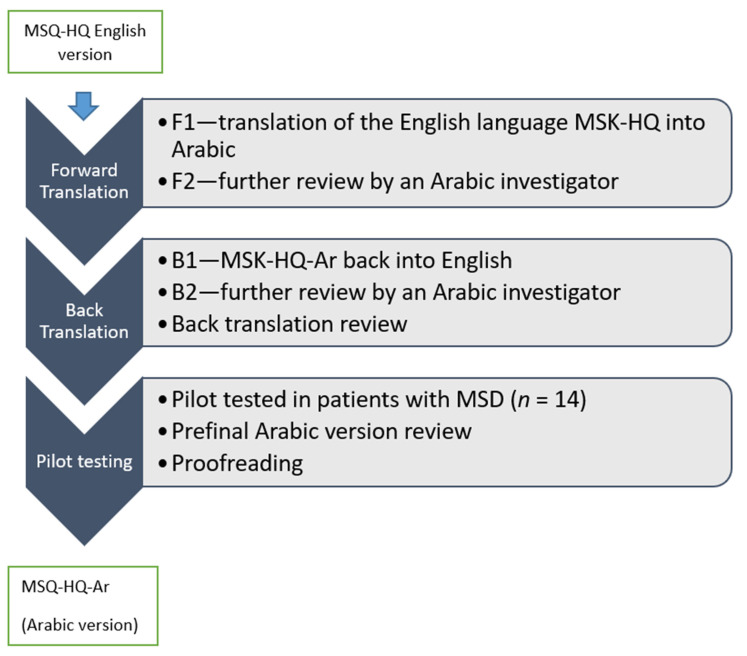
Flow chart for translation and cross-cultural adaptation of the Musculoskeletal Health Questionnaire into Arabic (MSK-HQ-Ar) from the English version. MSD, Musculoskeletal disorder.

**Table 1 ijerph-17-05168-t001:** Demographic data of the study participants (*n* = 149).

Age (Year)	*n* (%)
18–29	29 (19.5)
30–39	29 (19.5)
40–49	42 (28.1)
50–59	30 (20.1)
60–65	19 (12.8)
Mean (SD)	43.63 (13.69)
**Gender**	
Male	84 (56.4)
Female	64 (43.6)
**Level of Education**	
Primary	26 (17.4)
Intermediate	11 (7.4)
Secondary	37 (24.8)
University graduate	61 (40.9)
Postgraduate	8 (5.4)
**Employment Status**	
Student	19 (12.8)
Employee	60 (40.3)
Not working	54 (36.2)
Retired	16 (10.7)
**Marital Status**	
Married	109 (77.2)
Single	34 (22.8)
**BMI**	
Below 18.5	8 (5.3)
18.6–24.9	29 (19.4)
25–29.9	54 (36.3)
≤30	58 (39)
Mean (SD)	28.59 (5.601)

BMI, Body mass index; SD, standard deviation.

**Table 2 ijerph-17-05168-t002:** Clinical characteristics of the participants (*n* = 149).

MSD Region	*n* (%)
Spine	54 (36.2)
Upper extremity	27 (18.2)
Lower extremity	68 (45.6)
**Pain Site**	
Cervical	8 (5.4)
Lumber	45 (30.2)
Shoulder	25 (16.8)
Knee	62 (41.6)
Other (i.e., elbow, wrist, hip and ankle)	9 (6)
**Clinical Variables**	**Mean (±SD)**
NRS	5.52 (2.37)
MSK-HQ-Ar total score	32.29 (10.42)
EQ-5D-5L utility score	0.21 (0.22)

NRS, Numerical pain rating scale; MSK-HQ-Ar, Musculoskeletal Health Questionnaire; EQ-5D-5L, European quality of life five-dimension, five-level scale; MSD, Musculoskeletal disorder.

**Table 3 ijerph-17-05168-t003:** Descriptive statistics of the MSK-HQ-Ar scores (musculoskeletal disorders (MSDs) group).

Descriptive Statistics of the MSK-HQ-Ar Scores	Statistics	SE	
Mean	32.29	0.854	
Median	33.00		
Standard Deviation	10.428		
Range	48		
Skewness	−0.122	0.199	
Kurtosis	−0.189	0.395	
Normality test		df	Sig.
Shapiro-Wilk	0.998	149	0.226

SE, standard error; df, degrees of freedom; Sig, significance at *p* ≤ 0.05.

**Table 4 ijerph-17-05168-t004:** Statistics of the MSK-HQ-Ar items.

Item	Mean	SD (±)	Corrected Item to Total Correlation	Cronbach’s Alpha if Item Deleted
1. Pain/stiffness during the day	2.17	0.98	0.53	0.87
2. Pain/stiffness during the night	2.15	1.16	0.45	0.87
3. Walking	2.43	1.23	0.53	0.87
4. Washing/Dressing	3.07	1.25	0.50	0.87
5. Physical activity levels.	2.14	1.35	0.57	0.87
6. Work/daily routine	2.01	1.07	0.67	0.86
7. Social activities and hobbies	2.28	1.23	0.63	0.86
8. Needing help	2.42	1.19	0.55	0.87
9. Sleep	1.99	1.23	0.62	0.86
10. Fatigue or low energy	2.28	1.01	0.62	0.87
11. Emotional well-being	2.46	1.21	0.53	0.87
12. Understanding of your condition and any current treatment	2.50	1.33	0.41	0.88
13. Confidence in being able to manage your symptoms	2.46	1.23	0.45	0.87
14. Overall impact	1.87	1.06	0.64	0.86

**Table 5 ijerph-17-05168-t005:** Test-retest reliability of the MSK-HQ-Ar.

	Mean (SD)		ICC (95%CI)	SEM	MDC_95_
	Test Scores	Retest Scores			
**MSK-HQ-Ar total scores**	32.29 (10.4)	35.5 (10.4)	0.94 (0.92–0.95)	2.46	6.83

ICC: interclass correlation coefficient; CI: confidence interval; SEM: standard error of measurement; MDC_95_: minimal detectable change at 95% CI.

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
