# Peer review of "Cross-Cultural Adaptation and Validation of the Arabic Version of Musculoskeletal Health Questionnaire (MSK-HQ-Ar)"

_ijerph, 2020, doi:10.3390/ijerph17145168_

Round 1
Reviewer 1 Report
The cross-cultural adaptation and validation issues are not be mentioned in the Introduction Section. Why does the study focus on validation by reliability and ICC? There are two different topics in the study. Why the findings of the study are internal consistency, test-retest reliability, and construct validity in the Discussion Section? How to approve the construct validity?
Reviewer 2 Report
Thank you for the opportunity to review this manuscript. In this paper, the authors studied the Cross-cultural adaptation and validation of the Arabic version of Musculoskeletal Health Questionnaire (MSK-HQ-Ar). This study aimed to translate and culturally adapt the Musculoskeletal Health Questionnaire (MSK-HQ) into Arabic language (MSK- 76 HQ-Ar) and analyse its psychometric properties among patients with MSD in Saudi Arabia.
This cross sectional study used International Society for Pharmacoeconomics and Outcomes Research (ISPOR) guidelines to translate as well as validate the psychometric properties of MSK-HQ-Ar. The adapted MSK-HQ-Arabic version revealed acceptable psychometric properties and is a valid and reliable outcome measure to assess MSK health among Arabic speaking patients with MSD. The availability of such outcome measures in different languages may facilitate future comparisons among cross-cultural populations
The manuscript is interesting and I believe that this manuscript presents novelty for publication. Please, consider the following raised issues:
Line 71 – However, this scale is available only in English [17] and Italian [26] - the reference number is incorrect
Please add the conclusion section.

Reviewer 3 Report
Thank you for the opportunity to review the manuscript “Cross-cultural adaptation and validation of the Arabic version of Musculoskeletal Health Questionnaire (MSK-HQ-Ar)” for International Journal of Environmental Research and Public Health. This manuscript, using cross sectional data, offers (1) a translation of Musculoskeletal Health Questionnaire into Arabic, (2) an assessment of the validity and reliability of the aforementioned translation. This was an interesting manuscript, and I enjoyed reviewing it. There is much to like with this paper. Overall, it was thought provoking and enjoyable read. Importantly, the authors have provided an important means to assess musculoskeletal health among Arabic speaking patients who have been previously diagnosed with musculoskeletal disorders
Generally speaking, I have only favorable comments to offer. As such, I don’t want to slow down the review process asking for tangentially related revisions. So, instead, I will just say “well done.”
Round 2
Reviewer 1 Report
The topic of this study should contain the reliability issues.
In Line 221, "test-retest reliability" should be corrected.
Author Response
We agree with the reviewer and corrected the spelling mistake of 'test-retest reliability'.
Thanks for the point!